# Self-consistent deep approximation of retinal traits for robust and highly efficient vascular phenotyping of retinal colour fundus images

No Author Given

No Institute Given

**Abstract.** Retinal colour fundus images are a fast, low-cost, non-invasive way of imaging the retinal vasculature which could provide information about non-ocular, systemic health. Traditional approaches for retinal vascular phenotyping use handcrafted, multi-step pipelines that are computationally expensive and not robust to common quality issues. Recently, Deep Approximation of Retinal Traits (DART) was proposed which trains a neural network to mimic an existing pipeline in a more efficient and robust way. DART is orders of magnitude faster, more robust and repeatable. However, the original DART was not explicitly trained for repeatability, only provides a single retinal trait, Fractal Dimension (FD), and uses a limited set of augmentations. We propose DARTv2 that increases repeatability with a self-consistency loss, robustness with additional augmentations such as imaging overlays, and utility by adding Vessel Density (VD) as a second retinal trait in addition to FD. DARTv2 shows very high agreement (Pearson 0.9392 for FD and 0.9612 for VD, both $p << 0.05$) with AutoMorph, the pipeline it is based on. DARTv2 is far more robust than AutoMorph and also more robust than the original DART. Finally, DARTv2 is 200 times faster than AutoMorph and 4 times faster than the original DART, while taking up less storage space. DARTv2 will be made available to researchers upon publication.

**Keywords:** Retinal image analysis · Deep learning · Robustness

## 1 Introduction

Retinal colour fundus images are pictures of the retina, a layer of tissue at the back of our eyes that allows us to see. These images can be taken non-invasively in a few seconds with low-cost devices. They are crucial in ophthalmology for retinal disease screening, but also show the retinal vasculature in detail. The retinal blood vessels, in turn, could provide information about general vascular health and serve as a proxy for vascular changes elsewhere in the body, like the heart or the brain [11], a field of study also known as "oculomics" [16]. A common research paradigm is to extract retinal traits that summarise some aspect of the vasculature in a single number, e.g. Fractal Dimension (FD) which captures branching complexity of the blood vessels. Less complex retinal vasculature could indicate poorer vascular health, and indeed lower FD has been associated with

cardiovascular [15, 6, 18] and neurovascular [13, 10] disease. Vessel Density (VD), which captures how dense the vasculature is, likewise has shown associations with cardiovascular disease [18].

Retinal traits are traditionally extracted with handcrafted, multi-step pipelines that require high image quality like VAMPIRE [14] or AutoMorph [19]. In practice, a large share of images is excluded due to insufficient quality. In UK Biobank, a dataset collected specifically for research, 25-45% of the images are typically excluded [15, 12, 18]. These exclusions come at great cost: First, substantially reduced sample sizes and lower statistical power. Second, considerable selection bias as older, non-White, male, and less healthy subjects are more likely to be excluded [3], which exacerbates existing inequalities in healthcare research. Third, using these pipelines in clinical practice is virtually impossible if they fail in a quarter to half of the cases, and doubly so if they systematically fail more often for some subgroups.

Recently, Deep Approximation of Retinal Traits (DART) [4] was proposed to provide a more robust way of computing retinal traits. Follow-up work found DART to be substantially more repeatable than AutoMorph [1], surprisingly at any level of image quality exclusions, including in exclusively high-quality images. Thus, the DART paradigm does not only increase robustness but also repeatability in the absence of quality. A secondary, yet also important benefit is that DART is substantially faster than traditional pipelines, allowing to process images on low-end laptops.

However, the original version of DART had many drawbacks which we address in this work. First, the improved repeatability is only a lucky by-product of the increased robustness. Here, we propose a self-consistency loss to explicitly encourage repeatability. Second, DART used a limited set of data augmentations, and we also extend DART to VD in addition to FD.

## 2    Methods

### 2.1    Deep approximation of retinal traits

Briefly, DART approximates an existing pipeline with a neural network, which is trained to give the same output on high-quality images. However, during training, the model receives either original images or augmented versions that have their image quality synthetically degraded. Either way, DART needs to output the same number as the traditional pipeline did for the original, un-degraded image. This forces the model to ignore variations in image quality and instead extract all the available information about the retinal trait of interest. For example, shadows or pathology could obscure parts of the vessels. AutoMorph segments and skeletonises the vasculature, and then computes FD with box counting or VD with averaging, and would give a very low number if part of the vasculature is not segmented. A human clinician, on the other hand, would not be confused by the shadow and instead assess the part of the vasculature that is visible, which is what DART is designed to mimic. The original DART used VAMPIRE

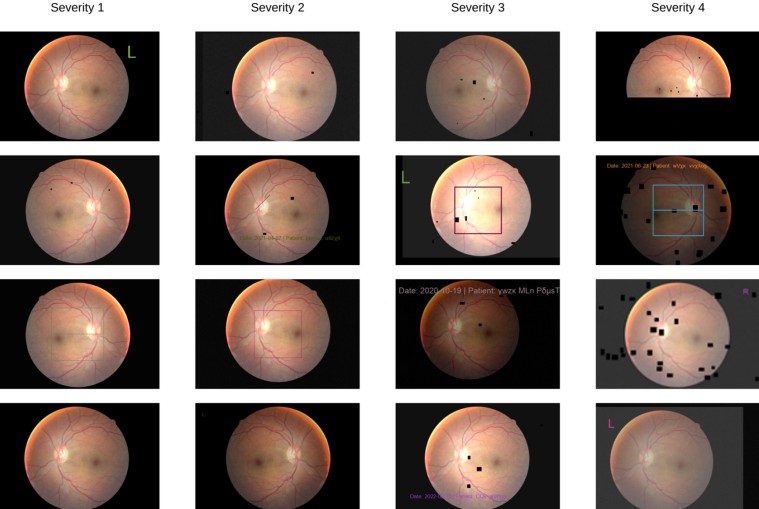

**Fig. 1.** Randomly sampled augmentations for each of the four levels of severity, using the same original image for illustration purposes.

[14] to generate ground-truths. In this work, we use AutoMorph [19] which is open-source and fully-automatic.

### 2.2 Augmentations

We define four levels of augmentation strength as shown in Fig. 1. These include horizontal flips, changing the brightness (lowest level $\pm 5\%$, highest $\pm 20\%$), contrast ($\pm 5\%$ to $\pm 60\%$), adding Gaussian blur and simulated imaging noise. We also include simple artefacts that remove multiple small parts of the image to simulate issues like dust or eyelashes, or parts of the images to simulate eyelids and partial shadows. These simulate common imaging issues. However, issues can also occur during the image export. Thus, we additionally simulate text overlays for laterality (left or right), dates and names, and grids indicating where an optical coherence tomography scan is taken. Finally, we simulate images being screenshots rather than proper exports by downsizing to a lower resolution and then back to our target resolution. While theoretically avoidable, in our experience, these are quite common in practice, and thus being robust to them is highly desirable.

### 2.3 Self-consistency loss

We explicitly encourage repeatability through a self-consistency loss which penalises poor repeatability across different augmentations. Concretely, during training in each mini-batch, we sample four augmented versions of each image - one of each level of severity - and obtain our model's predictions for each of

them. We then use normal mean squared error to penalise deviation from the value to original pipeline provided for the un-augmented image, but additionally also compute the standard deviation across the four versions of each image and add this to our loss. Thus, the model is trained to not only match the original pipeline but to output values that are self-consistent across different levels of image quality which should lead to increased robustness and repeatability in practice.

### 2.4   Increasing robustness through data filtering

For the original version of DART, despite 40% of UK Biobank having already been rejected by VAMPIRE, some poor quality images remained that provided noisy "ground-truths". For DARTv2, we thus aim to avoid these so the model does not replicate undesirable edge cases of the original pipeline. Thus, we filter using QuickQual's "Mega Minified Estimator" [2] which provides a one-dimensional, continuous quality score. QuickQual uses the same EyeQ image quality dataset [5] that AutoMorph's quality algorithm is trained on, but achieves state-of-the-art performance. Recent work found that the repeatability decreases beyond a QuickQual score of 0.8 [1], which indicates a 80% chance of being a bad image. This is about 2.5% of the training data, which we filter out. Note, we do not remove these images from the validation or test sets. Exploration of the training set revealed that there are still some extremely low values, presumably due to failures in the vessel segmentation. We thus clip the lower values of the targets in the training by setting the lowest one-thousandth of values to the 0.1-percentile.

### 2.5   DARTv2

We use a ConvNeXt [7], specifically the "femto" variant with an overlapping stem from the timm library [17], that was pre-trained on ImageNet. We add a small, perceptron as head with a single hidden layer with 512 hidden units and GELU activations. We normalize images using 0.5 as the mean and standard deviation parameters and resize to $256 \times 256$. The model is trained to minimise the sum of the mean squared error with the ground truth targets and consistency loss for 10 epochs with the AdamW [9] optimiser using mini-batches of 256 samples. Prior to the computation of the loss, predictions and targets are normalized to zero mean and unit variance using the training set statistics. We use a cosine learning rate schedule [8] with a linear warmup for the first two epochs and a single cosine cycle, a peak learning rate of $10^{-3}$, a weight decay of $10^{-2}$, clipping the maximum gradient norm to 0.1. We do not apply weight decay to the biases and initialise the final output layer to use the training set mean targets as output biases and zero weights.

### 2.6   Data

We use the EyePACS Diabetic Retinopathy dataset on Kaggle, which is openly available and consists of 88,702 colour fundus images acquired with a variety of

scanners, and use AutoMorph's FD and VD as ground-truths for training our model. We divided the dataset into train, validation, and test sets, allocating 76.5%, 8.5%, and 15% of the data, respectively. To ensure that each subject appeared only in one of the three sets, we split the data at the subject level. AutoMorph rejected 15.06% of the images due to insufficient image quality, and these images were excluded from further analysis. Thus, our training, validation and test sets contained 56,198, 6,245, and 10,952 images, respectively.

### 2.7    Evaluation

We quantify the agreement between AutoMorph and DARTv2 using the Pearson and Spearman correlation coefficients. Pearson is the most commonly used correlation and a linear measure. Spearman is a robust measure of correlation and is equivalent to computing the Pearson correlation of the ranks. Furthermore, we also fit a linear regression and report the best regression fit.

To compare the robustness of our model with the original DART and AutoMorph, we design a synthetic robustness test where images from the test are augmented and we then compare the agreement between each methods output for the original and the augmented image. A robust method should yield very similar values even in the face of augmentations, which would imply both greater robustness and repeatability in practice. While DARTv2 is trained with relatively strong augmentations, including text and OCT region overlays, it would be unfair to consider these as AutoMorph is not expected to be robust to those augmentations. Instead, we consider increasing and decreasing brightness 20% and contrast by 60%. These values were chosen as they visually change the images in a realistic way that is slightly but not overly challenging. In other words, in our opinion, a method for computing retinal traits should be fairly robust in the fact of these changes.

## 3    Results

### 3.1    Agreement on held-out test set

Fig. 2 shows the agreement between DARTv2 and AutoMorph on the original images from the held out test set. Generally, agreement is very high, with a Pearson correlation of 0.9392 for FD and 0.9612 for VD (all correlations are $p \ll 0.05$, as sample sizes are large). Spearman correlations are slightly lower but similar. The best regression fit indicates that the measures are very similar and can be interpreted in the same way. There are some outliers towards the bottom of the plot, where AutoMorph provides an extremely low value, whereas DARTv2 provides a low but not extremely low value. Manual inspection of some of these cases shows that AutoMorph struggles to segment the vasculature in these cases due to poor image quality or the presence of severe retinal pathology. Yet, these images had not been rejected by the AutoMorph quality scoring algorithm. We think that in these cases, AutoMorph outputs erroneously low values and it would be undesirable if DARTv2 replicated this behaviour.

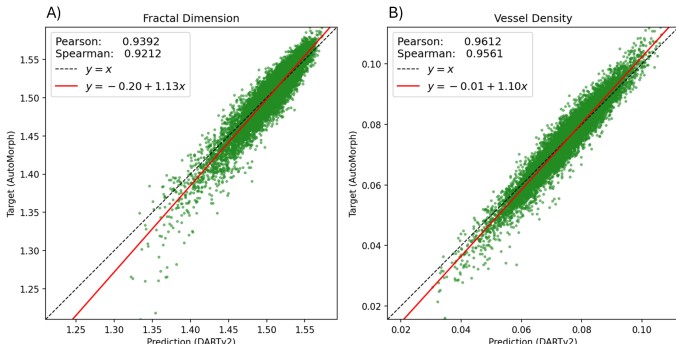

**Fig. 2.** Agreement between DARTv2 and AutoMorph on the held-out test set for A) Fractal Dimension and B) Vessel Density. The dashed black line indicates the identity line, the red line the best regression fit.

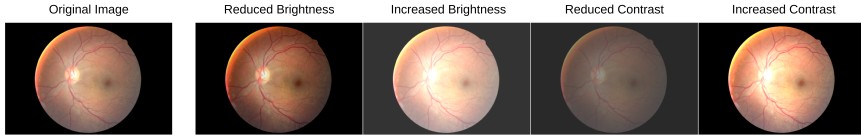

**Fig. 3.** Illustration of the augmentations used in our robustness testing using.

## 3.2   Robustness

Table 1 shows the results of the robustness evaluation. For both FD and VD, and for all considered augmentations, DARTv2 was substantially more repeatable than AutoMorph, demonstrating the advantage of a DART-based approach over traditional pipelines. DARTv2 was also more repeatable than the original DART, indicating that our approach improves the robustness of our model.

AutoMorph was unable to process up to 19% of the images depending on the type of augmentation due to numerical issues. The reported Pearson correlations for AutoMorph are excluding these values, which gives a more optimistic estimate of performance for AutoMorph, as difficult cases are the ones where processing fails. No cases, including those difficult cases, were excluded for the original

**Table 1.** Pearson correlation between the measurement on the original and augmented images for 1,000 randomly selected test set images. Higher is better, best result in bold. The original version of DART only outputs Fractal Dimension.

|  | Fractal Dimension | | | | Vessel Density | | | |
|---|---|---|---|---|---|---|---|---|
|  | +Brightness | -Brightness | +Contrast | -Contrast | +Brightness | -Brightness | +Contrast | -Contrast |
| AutoMorph [19] | 0.9731 | 0.6730 | 0.8348 | 0.4613 | 0.9794 | 0.7390 | 0.8714 | 0.5195 |
| DART (original) [4] | 0.9777 | 0.9335 | 0.9431 | **0.8611** | - | - | - | - |
| DARTv2 (ours) | **0.9961** | **0.9407** | **0.9775** | 0.8577 | **0.9971** | **0.9373** | **0.9844** | **0.8673** |
| Automorph fail rate | 14.10% | 1.50% | 1.50% | 19.00% | 13.90% | 0.10% | 1.10% | 13.60% |

**Table 2.** Inference speed and file sizes.

| | AutoMorph | AutoMorph (our optimisation) | DART (original) | DARTv2 (ours) |
|---|---|---|---|---|
| Images per second | 0.36 | 1.42 | 77.10 | **305.81** |
| Required disk space | 928MB | 928MB | 45 MB | **20MB** |

DART or our DARTv2. Thus, it is remarkable that despite this, DART and DARTv2 show substantially higher repeatability than AutoMorph. Indeed, the advantage of DARTv2 is smallest when brightness is increased, but this is after 14.1% and 13.9% of the images failed to be processed by AutoMorph.

### 3.3 Inference speed

Inference speed was measured on a desktop workstation with a last-gen high-end gaming GPU (Nvidia RTX 3090) and a four-year-old Intel i9 processor (i9-10900KF). To provide a maximally fair comparison, we measure performance by naively processing images sequentially rather than in batches, as implementing batch processing for AutoMorph is non-trivial while it would be easy to do for DART and DARTv2. Furthermore, we also optimise AutoMorph by removing all processing for retinal traits not considered in this study and by further parallelising non-GPU operations across multiple CPU cores where possible. This allows us to boost the speed of AutoMorph by almost four times.

Table 2 shows the results. DARTv2 is more than 800 times faster than AutoMorph and still 200 times faster than our optimised version. DARTv2 is also 4 times faster than the original DART, primarily due to using a smaller and more efficient model. In terms of filesize, DARTv2 is almost 50 times smaller than AutoMorph and less than half the size of the original DART. While even close to a GB of storage is not unreasonable nowadays, the smaller file sizes also mean faster downloads which will be especially beneficial for researchers without high-speed internet connections.

### 3.4 Effectiveness of our robustness-enhancing strategies

To evaluate the effectiveness of our robustness-enhancing strategies, we trained another DARTv2 model in the same way, except for removing the self-consistency loss and our augmentations. Fig. 4 shows the agreement of non-robust DARTv2 with AutoMorph on the test set. As expected, agreement is substantially higher when not encouraging robustness as the model is able to learn the behaviour of the original pipeline in edge cases as well, leading to better agreement. However, our goal is not to match the original pipeline perfectly but instead only learn the mimic its consistent behaviour that captures a meaningful aspect of the vasculature. When comparing the robustness of the proposed DARTv2 and the non-robust version (Table 3), we indeed find that DARTv2 is more robust for each of the eight comparisons, indicating the effectiveness of our robustness strategies.

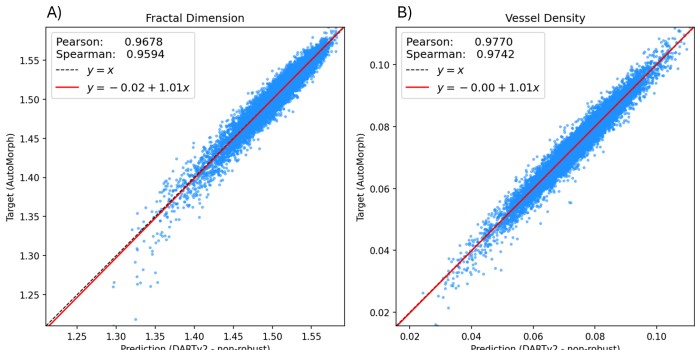

**Fig. 4.** Agreement between the non-robust version of DARTv2 and AutoMorph on the held-out test set for A) Fractal Dimension and B) Vessel Density. The dashed black line indicates the identity line, the red line the best regression fit.

**Table 3.** Pearson correlation between the measurement on the original and augmented images for 1,000 randomly selected test set images for our proposed DARTv2 and the non-robust version of DARTv2. Higher is better, best result in bold.

|  | Fractal Dimension | | | | Vessel Density | | | |
|---|---|---|---|---|---|---|---|---|
|  | +Brightness | -Brightness | +Contrast | -Contrast | +Brightness | -Brightness | +Contrast | -Contrast |
| DARTv2 (ours) | **0.9961** | **0.9407** | **0.9775** | **0.8577** | **0.9971** | **0.9373** | **0.9844** | **0.8673** |
| DARTv2 - no robustness | 0.9750 | 0.9182 | 0.9587 | 0.8251 | 0.9801 | 0.9202 | 0.9660 | 0.8472 |

## 4    Conclusion

We presented DARTv2, an improved model for deep approximation of retinal traits with increased robustness and self-consistency. Our experiments show that DARTv2 not only has very good agreement with AutoMorph on the original images while being substantially more robust, but it is also more robust than the original DART model. Furthermore, DARTv2 is more than 800 times faster than AutoMorph and 4 times faster than the original DART. Our experiments show that our self-consistency loss and augmentation strategies indeed improve robustness. We hope that DARTv2's robustness will allow researchers to exclude fewer images, which would also partially alleviate the selection bias and unfairness introduced by these exclusions. The increased efficiency of DARTv2 could help democratise retinal image analysis.

Future work should expand on the self-consistency loss proposed here and investigate additional strategies for encouraging DART-style models to learn desirable properties. While we expanded on the augmentations used in the original DART, additional augmentations such as simulating the magnification effect due to variations in refractive error should be investigated. Finally, in the future additional retinal traits like tortuosity could be added as well as image quality scoring, so researchers can use a single model instead of using DARTv2 and QuickQual separately.

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
