# OpenReview forum: "Self-consistent deep approximation of retinal traits for robust and highly efficient vascular phenotyping of retinal colour fundus images"
_MICCAI.org/2024/Workshop/MSB — MICCAI Student Board EMERGE Workshop 2024 Oral_

### Official Review · Reviewer_2m62 · 2024-07-05

**Recommendation:** 4
**Confidence:** 4

**Clarity:**

The paper is clear and well-written, with minor areas for improvement in clarity

**Feedback:**

- In the last paragraph of the introduction it is stated that DART's improved repeatability is only a lucky by-product. I would re-write that sentence and rething what are the factors in the original DART method that could have lead to that outcome. Protentially repeatability and robustness are more related than expected by the authors.
- The citation to the EyePACS dataset is missing.
- In the discussion or future work I would mention the limitation of having to use AutoMorph (or another method like VAMPIRE) as ground truth for training.

**Justification:**

The paper is well-written and the results show some improvement over DART.

**Reproducibility:**

Sufficient amount of details available for reproducing the main results, and open access is provided (or promised upon acceptance) to source code and/or data

**Strengths:**

- The paper is well-written and easy-to-follow.
- The motivation for the self-consistency loss is clear.
- Predicting an additional retinal trait besides Fractal Dimension can increase the clinical utility of the method.

**Summary:**

The paper extends DART, a method that predicts the retinal trait of Fractal Dimension, to DARTv2, which is trained with extended data augmentations, using a different method to generate the ground truth, to predict the additional trait of Vessel Density using a self-consistency loss between an image and its augmentated versions. The method shows promising results when compared against non-deep-learning methods and DART.

**Weaknesses:**

- What are the exclusion criteria for the images with insufficient quality? Could you provide some visual examples of images that passed and images that failed the quality check?
- Since a benefit and goal of the approach is improved repeatability I would define that term in the introduction, as it could have multiple meanings in different settings. And how is repeatability different than robustness regarding this particular task?
- In Section 2.2, which of the augmentations are proposed specifically for DARTv2 and which ones were already used in DART?
- The fact that the method uses AutoMorph as ground truth limits its upper bound performance, especially since there are many drawbacks associated with AutoMorph, as discussed in the Introduction and Methods sections of the paper. Why was AuthMorph selected over VAMPIRE, for example?
- The self-consistency loss, from my understanding, operates on the model predictions. Have the authors considered using a metric-learning style of loss, where the features of the images and their augmentations are pulled together in the embedding space? (similar to SimCLR)
- The robustness experiments would be more interesting if the images used as "difficult" samples were not synthetic ones with increased brightness, but real images that had low image quality scores. The real images could be clustered based on their image quality score and the metrics could be reported for each image quality cluster.
- It is counterintuitive that in the Methodology it is motivated that the additional data augmentations are needed and useful, but for the experiments it is stated that "a method for computing retinal traits should be fairly robust..." and ".. in a way that is realistic but not overly challenging.". The notions of "fairly robust" and "not overly challenging" should be clarified, and preferably quantified. Why do we need extra augmentation if it is hypothesized that these might lead to very challenging images that are not expected in real life? I would rephrase these sentences.
- Are AutoMorph and its optimized version running on the CPU or on the GPU?

---

> ### Author Response · Authors · 2024-07-13
> **Rebuttal by Authors**
>
> Dear Reviewer,
>
> Thank you for your thorough review and constructive feedback on our DARTv2 paper. We appreciate your recognition of the paper's strengths and clarity. We will address your concerns and suggestions as follows:
> - Data filtering: AutoMorph's built-in quality assessment rejected 15.06% of the images due to insufficient quality. These images were excluded from further analysis in all sets (training, validation, and test). For DARTv2 training, we applied an additional filter using QuickQual scores. Images with a QuickQual score above 0.8 (indicating an 80% chance of being a bad image) were removed from the training set only. This affected approximately 2.5% of the training data. We retained these images in the validation and test sets to maintain a realistic evaluation scenario.
> - Repeatability and robustness: In this context, repeatability refers to the consistency of retinal trait measurements across multiple images of the same eye, even with variations in image quality or acquisition conditions. Robustness refers to the model's ability to maintain accurate predictions across a wide range of image qualities and variations. This includes performance on images with artifacts, different levels of brightness or contrast, or even partial occlusions.
> - Augmentations: Thank you for highlighting this. We agree that distinguishing between DART and DARTv2 augmentations would improve clarity. Specifically, DARTv1 included (horizontal (p = 0.5) and vertical flip (p = 0.1), mild affine transformations (p = 0.15, rotation by up to ±10°, shear of up to ±5°, and scaling by ±5%)), and one of three artifacts: many smaller rectangular holes placed across the retina, b) a single large square hole, or c) "chop" off the bottom or top part of the image. In contrast, DARTv2 presents much more extensive data augmentation, adding realistic GUI components such as text, letter representing the side, dust and ROI marker, all with p=0.1, before applying one of the four levels of severity defined as:
>   - Severity 1:
>     - Horizontal flip (p = 0.5)
>     - ± 5% Brightness and Contrast
>   - Severity 2:
>     - Severity 1
>     - Gaussian blur (p = 0.25)
>     - Shift of up to 5% (p = 0.5)
>     - ISO noise, color shift of ± 5% (p = 0.5)
>     - Coarse dropout, 3 max holes, small size (p = 0.25)
>   - Severity 3:
>     - Severity 2
>     - ± 15% Brightness
>     - ± 40% contrast
>     - Gaussian blur (p = 0.5)
>     - Up to 5% scale transform (p = 0.5)
>     - Coarse dropout, 3 max holes, medium size (p = 0.25)
>     - Downsize and upsize, 60% - 80% of original resolution (p=0.1)
>   - Severity 4:
>     - Severity 3
>     - Brightness variations of ± 20%
>     - Contrast variations of ± 60%
>     - Shift of up to 10% (p = 0.5)
>     - Scale transform of up to 10% (p = 0.5)
>     - Downsize and upsize, 20% - 80% of original resolution (p=0.1)
>     - One of:
>       - Coarse dropout, 3 max holes, big size
>       - Cutout, up to 10 holes
>       - Removing up to 50% of the image, starting from a random side
> - AutoMorph as ground truth: We appreciate your insight on this limitation. AutoMorph was chosen for its accessibility and open-source nature. We acknowledge that this choice may impact DARTv2's upper-bound performance, and we are aware of its limitations. Comparing our results with the ones obtained using VAMPIRE or even using multiple ground truth sources could be an interesting future line of work.
> - Self-consistency loss: Thank you for suggesting a metric-learning style loss. This is an intriguing idea that could potentially improve feature representation. We will consider exploring this approach in future work.
> - Robustness experiments: We appreciate your suggestion to use real low-quality images for a more realistic assessment of model performance. This approach would indeed provide valuable insights into DARTv2's real-world robustness. However, it presents challenges, as it would require expert validation of the ground truth generated by AutoMorph or an alternative method for these low-quality images, as the segmentation step could fail.
> - Augmentation motivation: Thank you for this observation. We will rephrase these sentences to better align with our motivation. Our augmentations are indeed designed to mimic real-world scenarios, with different severity levels to prepare the model for diverse conditions while maintaining practicality.
> - AutoMorph implementation: To clarify, both AutoMorph and its optimized version were run on the same GPU (Nvidia RTX 3090) used for DARTv2 training and inference, ensuring a fair comparison of processing speeds.
>
> We appreciate your recommendation and confidence in our work. These clarifications and considerations will help improve the paper's clarity and provide good pointers for future work.
>
> Thank you again for your valuable insights and suggestions.
>
> Sincerely,
>
> The Authors

---

### Official Review · Reviewer_qJYM · 2024-07-07

**Recommendation:** 4
**Confidence:** 4

**Clarity:**

The paper is generally clear but has some clarity issues that could be addressed with moderate revision

**Feedback:**

1. Consider moving the content related to DART (section 2.1) to the introduction or related work section for better flow.

2. It would be beneficial to illustrate the DARTv2 pipeline at the beginning to provide an overview of the proposed method, including the framework backbone and a brief introduction to the robustness-enhancing strategies.

3. Separate the details of the proposed framework into different sections, such as the proposed loss function, augmentation techniques, and data filtering methods, following the DARTv2 overview.

4.   Implementation details demonstrated in section 2.5 should be demonstrated in the Experiment and Results section.

5. Consider including the evaluation metrics as part of the implementation details and placing them in the Experiment and Results section rather than the Method section.

**Justification:**

The paper makes several valuable contributions, such as improving the robustness of the existing DART framework while reducing the number of parameters and required storage. However, some technical details are missing, such as the formula for the proposed loss function and a diagram illustrating the proposed method, which makes the work difficult to reproduce. Additionally, the paper is not structured in a systematic manner, making it challenging to follow.

**Reproducibility:**

Some amount of details available for reproducing the main results, and open access details are unclear

**Strengths:**

1. The self-consistency loss proposed in this work not only matches the original pipeline but also ensures output values are self-consistent across different levels of image quality, thereby improving repeatability.

2. The inclusion of an additional data augmentation technique, specifically text overlays, enhances the robustness of the framework.

3. The proposed method allows for the evaluation of another phenotype, vessel density, in addition to those assessed by DART.

4. The method requires fewer training parameters and less storage space compared to existing approaches.

**Summary:**

Retinal colour fundus images are a fast, low-cost, non-invasive way of imaging the retinal vasculature, which can provide valuable information about systemic health beyond ocular conditions. In this paper, the authors propose DART v2 to generate retinal vascular phenotypes from retinal images, building on the existing DART approach. The self-consistency loss is introduced to enhance the framework's repeatability. Additional augmentations, such as imaging overlays, are incorporated to improve robustness. Alongside Fractal Dimension (FD), Vessel Density (VD) is also considered as a second retinal trait to increase the method's utility.

**Weaknesses:**

1. Although the self-consistency loss is elaborated in the text, the specific loss function is not provided in the paper, making it difficult to understand how it operates during model training.

2. The proposed pipeline lacks a diagram or figure to illustrate the input, output, and network structure, complicating the reproducibility of the work.

4. Section 3.4 presents results generated using two robustness-enhancing strategies, but the effectiveness of each strategy is not individually evaluated, making it challenging to assess their contributions.

---

> ### Author Response · Authors · 2024-07-13
> **Rebuttal by Authors**
>
> Dear Reviewer,
>
> Thank you for your thorough review and constructive feedback on our DARTv2 paper. We appreciate your recognition of our contributions and the strengths you have highlighted. We'll address your concerns and suggestions to improve the paper's clarity and reproducibility.
> Addressing your key points:
>
> - Self-consistency loss: We will include the mathematical formulation in the revised manuscript:
>   $$
>   \mathcal{L}\_{\text{self-consistency}} = \frac{1}{B} \sum\_{i=1}^{B} \left( \frac{1}{M} \sum\_{j=1}^{M} \text{Var}\left[ f(S_j(x_i)) \right] \right)
>   $$
> Where $B$ is batch size, $M$ is augmentation severity levels, $f$ is the neural network, and $S_j(x_i)$ represents the augmented version of image $x_i$.
> - Pipeline diagram: We will add a figure illustrating the DARTv2 pipeline, including input, output, and network structure, to enhance clarity and reproducibility.
> - Regarding individual ablations, we provide a combined ablation of augmentation and self-consistency loss in Table 3. The original DART already utilized augmentations, including those we use for robustness testing (brightness and contrast). Therefore, we focused on the self-consistency loss, which requires augmentations to compute the standard deviation across multiple augmented versions of the same image. While we could have included a version with augmentations alone, we believe the comparison with the original DART adequately covers this aspect. Consequently, the suggested experiment would not significantly alter the interpretation of our results. However, we will consider this for future work.
> - We greatly appreciate your specific feedback on reorganizing the paper structure to enhance clarity and readability. We will implement these suggestions in the final manuscript.
> - To aid reproducibility and facilitate the use of DARTv2 by the research community, we will release the training and inference code on GitHub upon paper acceptance.
>
> Thank you for your recommendations and confidence in our work. We believe these revisions will significantly improve the paper's clarity, structure, and reproducibility.
>
> Sincerely,
>
> The Authors

---

### Official Review · Reviewer_iL4B · 2024-07-11

**Recommendation:** 4
**Confidence:** 4

**Clarity:**

The paper is generally clear but has some clarity issues that could be addressed with moderate revision

**Feedback:**

-  I did not well understand why "the improved repeatability is only a lucky by-product of the increased robustness". Maybe this idea should be rephrased/better explained.
- I would mention that there might still be some exclusion bias and clearly explain how you tackled this problem. It seems like many steps have been taken to overcome this issue, but a small paragraph could help clarify further this important aspect, i.e. what are the potential biases introduced by the exclusion of low-quality images during training?
- The self-consistency loss is an important novelty and could be explained in more depth. A more detailed mathematical formulation/possibly a visual representation of how this loss contributes to the training process would clarify its role and importance.

**Justification:**

The paper introduces a new model that outperforms its predecessor. While some parts could be better written, explained and presented, the authors have performed a thorough evaluation of their newly introduced method. Overall, DARTv2 presents significant advancements in terms of repeatability, robustness, and efficiency.

**Reproducibility:**

Sufficient amount of details available for reproducing the main results, but open access is not provided to source code and/or data

**Strengths:**

- Experiments and results are presented very thoroughly. In particular, the evaluation is very comprehensive.
- Model includes both FD and VD for better assessment of retinal vascular health.
- Multiple robustness techniques that could be encountered have been considered: brightness, contrast, blur, removing certain parts, etc...
- Higher performance of DARTv2 compared to DART.
- Much more efficient model compared to the previous version: faster and smaller in size.

**Summary:**

The paper proposes DARTv2, an improved version of DART for analyzing retinal color fundus images. DARTv2 aims to overcome the limitations of traditional retinal vascular phenotyping methods that are computationally expensive and sensitive to image quality issues. The improvements in DARTv2 include the introduction of a self-consistency loss to enhance repeatability, the inclusion of VD alongside FD, and the use of additional augmentations to improve robustness. DARTv2 demonstrates high agreement with AutoMorph. In particular, it shows a significantly enhanced robustness, faster processing speeds, and reduced storage.

**Weaknesses:**

- One of the main weakness is the fact the approach might showcase some exclusion bias. By depending on AutoMorph to "select" the images and their ground truths, the authors might have introduced some bias into their model and therefore, their results could be debated. Even though there has been a thorough effort to combat this issue, this exclusion bias might still be present.
- The paper only introduces and leverages one dataset. It could have been very valuable and interesting to evaluate the method on another dataset.
- A diagram of the model/pipeline architecture could be a great addition.
- It is unclear if the augmentation techniques introduced in this paper were new or reused from DART.
- Some parts of the abstract and introduction lack clarity and be better phrased.

---

> ### Author Response · Authors · 2024-07-13
> **Rebuttal by Authors**
>
> Dear Reviewer,
>
> We sincerely appreciate your thorough review and constructive feedback. We're pleased you found our evaluation comprehensive and recognized DARTv2's strengths in robustness, efficiency, and inclusion of both FD and VD.
>
> Addressing your key points:
>
> - Exclusion bias: We acknowledge this concern and will expand our discussion on potential biases and mitigation strategies in the revised manuscript.
> - Single dataset: While EyePACS offers significant variability (88,702 images, multiple cameras, diverse ethnicities, healthy and diseased retinas), we agree that evaluating on additional datasets is crucial for establishing generalizability. We'll include this as a key direction for future work.
> - Model architecture diagram: We'll add this to improve clarity.
> - Augmentation techniques: We'll clarify the differences between DARTv1 and DARTv2 augmentations. Specifically, DARTv1 included (horizontal (p = 0.5) and vertical flip (p = 0.1), mild affine transformations (p = 0.15, rotation by up to ±10°, shear of up to ±5°, and scaling by ±5%)), and one of three artifacts: many smaller rectangular holes placed across the retina, b) a single large square hole, or c) "chop" off the bottom or top part of the image. In contrast, DARTv2 presents much more extensive data augmentation, adding realistic GUI components such as text, letter representing the side, dust and ROI marker, all with p=0.1, before applying one of the four levels of severity defined as:
>
>   - Severity 1:
>     - Horizontal flip (p = 0.5)
>     - ± 5% Brightness and Contrast
>   - Severity 2:
>     - Severity 1
>     - Gaussian blur (p = 0.25)
>     - Shift of up to 5% (p = 0.5)
>     - ISO noise, color shift of ± 5% (p = 0.5)
>     - Coarse dropout, 3 max holes, small size (p = 0.25)
>   - Severity 3:
>     - Severity 2
>     - ± 15% Brightness
>     - ± 40% contrast
>     - Gaussian blur (p = 0.5)
>     - Up to 5% scale transform (p = 0.5)
>     - Coarse dropout, 3 max holes, medium size (p = 0.25)
>     - Downsize and upsize, 60% - 80% of original resolution (p=0.1)
>   - Severity 4:
>     - Severity 3
>     - Brightness variations of ± 20%
>     - Contrast variations of ± 60%
>     - Shift of up to 10% (p = 0.5)
>     - Scale transform of up to 10% (p = 0.5)
>     - Downsize and upsize, 20% - 80% of original resolution (p=0.1)
>     - One of:
>       - Coarse dropout, 3 max holes, big size
>       - Cutout, up to 10 holes
>       - Removing up to 50% of the image, starting from a random side
> - Abstract and introduction clarity: We'll revise these sections for improved clarity.
>
> Regarding specific feedback:
>
> - We'll rephrase the statement about improved repeatability for clarity.
> - We'll add a detailed explanation of the self-consistency loss, including its mathematical formulation:
> Given a batch of images $\\{x_i\\}\_{i=1}^B$, where each $x_i$ represents an individual image and $B$ denotes the batch size, our self-consistency loss aims to enforce consistent predictions by the neural network $f$ on augmented versions of these images. For each image in the batch, we apply a set of augmentation functions $\\{S_j\\}\_{j=1}^M$, each representing a different level of augmentation severity. The loss $\mathcal{L}\_{\text{self-consistency}}$ is computed as the average variance of the neural network's predictions over these augmented images, normalized across the batch. This encourages the model to produce stable outputs for variations introduced by the augmentations, thereby improving the robustness of the model against input perturbations. The mathematical formulation of the self-consistency loss is given by:
>
>   $$
>   \mathcal{L}\_{\text{self-consistency}} = \frac{1}{B} \sum\_{i=1}^{B} \left( \frac{1}{M} \sum\_{j=1}^{M} \text{Var}\left[ f(S_j(x_i)) \right] \right)
>   $$
>
>
>   where:
>
>   - $B$ is the batch size.
>   - $M$ is the number of different levels of severity for the augmentation functions.
>   - $f$ represents the neural network.
>   - $S_j(x_i)$ represents the augmented version of the image $x_i$ with the $j$-th level of severity.
>   - $\text{Var}[\cdot]$ denotes the variance across the network's outputs for the augmented versions of the same image, encouraging the model to have a consistent output across different augmentations of the same image.
>
> We're grateful for your recommendation and will carefully revise the manuscript to address your valuable feedback.
> Sincerely,
>
> The Authors

---

### Meta-Review · Area_Chair_2cLi · 2024-07-16

**Recommendation:** Accept (Oral)
**Confidence:** 4

**Metareview:**

This paper proposes DARTv2, a method for analyzing retinal color fundus images. DARTv2 builds upon the previous DART model and shows some promise. The reviewers have mentioned areas where the paper can be strengthened to improve clarity and methodological rigor. The reviewers recommended using another dataset and real low-quality images to strengthen the claims about the method's performance and generalizability. The authors have sufficiently addressed the reviewers’ comments. Including a model architecture diagram, clarifying definitions and technical details and discussing limitations and future directions in the final version would enhance clarity. The authors also promised to release the source code.

---

### Decision · Program_Chairs · 2024-07-16

Accept (Oral)